# Gut microbiome diversity is an independent predictor of survival in cervical cancer patients receiving chemoradiation

Travis T. Sims [1,7], Molly B. El Alam [2,7], Tatiana V. Karpinets[3], Stephanie Dorta-Estremera[4], Venkatesh L. Hegde[4], Sita Nookala[4], Kyoko Yoshida-Court[2], Xiaogang Wu[3], Greyson W. G. Biegert[2], Andrea Y. Delgado Medrano[2], Travis Solley[2], Mustapha Ahmed-Kaddar[2], Bhavana V. Chapman[2], K. Jagannadha Sastry[4], Melissa P. Mezzari [5], Joseph F. Petrosino[5], Lilie L. Lin [2], Lois Ramondetta[1], Anuja Jhingran[2], Kathleen M. Schmeler[1], Nadim J. Ajami[3], Jennifer Wargo [6], Lauren E. Colbert[2✉] & Ann H. Klopp[2✉]

Diversity of the gut microbiome is associated with higher response rates for cancer patients receiving immunotherapy but has not been investigated in patients receiving radiation therapy. Additionally, current studies investigating the gut microbiome and outcomes in cancer patients may not have adjusted for established risk factors. Here, we sought to determine if diversity and composition of the gut microbiome was independently associated with survival in cervical cancer patients receiving chemoradiation. Our study demonstrates that the diversity of gut microbiota is associated with a favorable response to chemoradiation. Additionally, compositional variation among patients correlated with short term and long-term survival. Short term survivor fecal samples were significantly enriched in *Porphyromonas*, *Porphyromonadaceae*, and *Dialister*, whereas long term survivor samples were significantly enriched in *Escherichia Shigella*, *Enterobacteriaceae*, and *Enterobacteriales*. Moreover, analysis of immune cells from cervical tumor brush samples by flow cytometry revealed that patients with a high microbiome diversity had increased tumor infiltration of CD4+ lymphocytes as well as activated subsets of CD4 cells expressing ki67+ and CD69+ over the course of radiation therapy. Modulation of the gut microbiota before chemoradiation might provide an alternative way to enhance treatment efficacy and improve treatment outcomes in cervical cancer patients.

[1] Department of Gynecologic Oncology and Reproductive Medicine, The University of Texas MD Anderson Cancer Center, Houston, TX, USA. [2] Department of Radiation Oncology, The University of Texas MD Anderson Cancer Center, Houston, TX, USA. [3] Department of Genomic Medicine, The University of Texas MD Anderson Cancer Center, Houston, TX, USA. [4] Department of Thoracic Head and Neck Medical Oncology, The University of Texas MD Anderson Cancer Center and the UTHealth Graduate School of Biomedical Sciences at Houston, Houston, TX, USA. [5] Department of Molecular Virology and Microbiology, Alkek Center for Metagenomics and Microbiome Research, Baylor College of Medicine, Houston, TX, USA. [6] Department of Surgical Oncology, The University of Texas MD Anderson Cancer Center, Houston, TX, USA. [7] These authors contributed equally: Travis T. Sims, Molly B. El Alam. ✉email: lcolbert@mdanderson.org; aklopp@mdanderson.org

Cervical cancer (CC) continues to be one of the leading causes of cancer-associated mortality globally[1]. More than 500,000 new cases of invasive CC will be diagnosed worldwide in 2020, resulting in over 300,000 deaths[2]. Multi-modality therapy consisting of definitive chemoradiation (CRT) comprising external-beam radiotherapy (EBRT) followed by intracavitary brachytherapy with concurrent systemic chemotherapy continues to be the standard of care in clinical practice for locally advanced disease[3].

The fecal or gut microbiome, a diverse community of bacteria, archaea, fungi, protozoa, and viruses, is thought to influence host immunity by modulating multiple immunologic pathways, thus impacting health and disease[4–6]. The diversity of the gut microbiome is defined as the number and relative abundance distribution of these distinct types of microorganisms colonizing within the gut[7]. Studies have suggested that dysbiosis of the gut microbiome confers a predisposition to certain malignancies and influences the body's response to a variety of cancer therapies, including chemotherapy, radiotherapy, and immunotherapy[6,8–11]. For example, melanoma patients are more likely to have a favorable response to immune checkpoint blockade and exhibit improved systemic and antitumor immunity if they have a more diverse intestinal microbiome[11].

Radiotherapy promotes the activation of T cells directed against tumor antigens[12–15]. In combination with immunotherapy, radiotherapy can maximize the antitumor immune response and promote durable disease control[16,17]. We theorize that the gut microbiota may modulate radioresponse through immunologic mechanisms[14,18]. Limited studies have examined the association between cancer treatment and the gut microbiome in patients with gynecological malignancies. Our collaborative group has previously studied the longitudinal effects CRT has on the gut microbiome. Our analysis found that CRT induces a decline in gut diversity by the fifth week of treatment. After CRT, gut microbiome diversity tends to return to baseline levels, but its structure and composition remain significantly altered[19]. CRT has also been shown to alter overall richness and diversity of the gut microbiome in cancer patients[20,21]. The mechanism by which this occurs has not been fully elucidated, but cumulative fractions of CRT may induce outgrowth of radioresistant microbial taxa, ultimately selecting bacteria tolerant to radiation-induced insults to persist.

Studies investigating the gut microbiome and outcomes in cancer patients often do not adjust for confounding patient and tumor characteristics. To assess this, we sought to identify independent gut microbial risk factors in CC patients receiving CRT and to evaluate their impact on survival. We hypothesize that gut microbial differences may affect clinical outcomes in patients with CC.

## Results

**Patient Characteristics.** A total of 55 patients with a mean age of 47 years (range, 29–72 years) volunteered to participate in this study. The patients received standard treatment for CC with 5 weeks of EBRT and weekly cisplatin. After completion of EBRT, patients received brachytherapy. For evaluation of treatment response, patients underwent magnetic resonance imaging (MRI) at baseline and week 5 and positron emission tomography/computed tomography 3 months after treatment completion (Fig. 1a). Most patients had stage IIB disease (51%) and squamous histology (78%). Their clinicopathologic data are summarized in Supplementary Table 1 and 4. We staged CC using the 2014 International Federation of Gynecology and Obstetrics (FIGO) staging system. The median cervical tumor size according to MRI was 5.4 cm (range, 1.2–11.5 cm). Thirty patients (55%) had lymph node involvement according to imaging studies. We first analyzed the bacterial 16 S rRNA (16 Sv4) fecal microbiota at baseline with respect to disease histology, grade, and stage. We found that the baseline α-diversity (within tumor samples) and β-diversity (between samples) of the fecal microbiome in the CC patients did not differ according to histology, grade, or stage ($P > 0.05$) (Supplementary Fig. 1a–f).

**Univariate and multivariate analysis of factors affecting recurrence-free survival (RFS) and overall survival (OS).** In the univariate Cox proportional hazard regression model predicting RFS, three covariates showed $P < 0.1$. As shown in Table 1, univariate analysis identified older age (Hazard Ratio (HR) of 0.93 (95% CI = 0.87–0.98, $P = 0.0096$)), Shannon diversity index (SDI) (HR of 0.51 (95% CI = 0.23–1.1, $P = 0.087$)) and body mass index (BMI) (HR of 0.92 (95% CI = 0.84–1, $P = 0.096$)) as risk factors for RFS. Multivariate survival analyses identified BMI and SDI as independent prognostic factors for RFS with a HR of 0.87 (95% CI = 0.77–0.98, $P = 0.02$) and 0.36 (95% CI = 0.15–0.84, $P = 0.018$) respectively. As shown in Table 2, univariate analysis identified SDI (HR of 0.34 (95% CI = 0.1–1.1, $P = 0.08$) and BMI (HR of 0.83 (95% CI = 0.69–1, $P = 0.055$)) as risk factors for OS. For OS, multivariate survival analyses again identified BMI and SDI as independent prognostic factors with an HR of 0.78 (95% CI = 0.623–0.97, $P = 0.025$) and 0.19 (95% CI = 0.043–0.83, $P = 0.028$), respectively. Univariate Cox regression analysis for RFS and OS survival with respect to alpha diversity all time points is shown in Supplementary Table 2 and 3.

**Baseline gut microbiota diversity is associated with favorable responses.** During the median follow-up period of 24.5 months, seven patients died; all patients (12.7% of the total study population) died of disease. Figure 1 shows the Kaplan–Meier curves for RFS and OS. Given our univariate and multivariate analyses performed by Cox proportional hazard model identified SDI as an independent predictor for RFS and OS, we first tested the relationship between gut diversity and RFS and OS in our cohort by stratifying patients based on high and low SDI. We stratified the patients by SDI as high-diversity versus low-diversity groups based on the cutoff value of SDI (2.69) calculated by receiver operating characteristic curve (ROC). The median RFS was 44 months for patients with low fecal alpha diversity at baseline, and not reached for patients with high fecal alpha diversity at baseline ($P = 0.16$) (Fig. 1a). The median OS was 48 months for patients with low fecal alpha diversity at baseline, and again not reached for patients with high fecal alpha diversity at baseline ($P = 0.094$) (Fig. 1b, Supplemental Fig. 3). Next, because our univariate and multivariate analyses performed by Cox proportional hazard model also identified BMI as an independent predictor for RFS and OS, we tested the relationship between diversity and RFS and OS in our cohort by stratifying patients based on high and low Shannon diversity metric and normal or high BMI. As shown in Fig. 1d, e, when BMI and gut diversity are stratified for at baseline, patients with normal BMI and higher SDI had a longer median RFS duration ($P = 0.0023$) (Fig. 1d). OS (Fig. 1e) was longer for patients with normal BMI and higher gut diversity ($P = 0.19$).

**Compositional difference in gut microbiome in response to CRT.** To further investigate whether the composition of gut microbiome was associated with response to CRT, we used linear discriminant analysis (LDA) Effect Size analysis to identify bacterial taxa that were differentially enriched in short-term and long-term CC patients ($P < 0.05$; LDA score >3.5). In all patients, multiple taxa differed significantly at baseline between short and

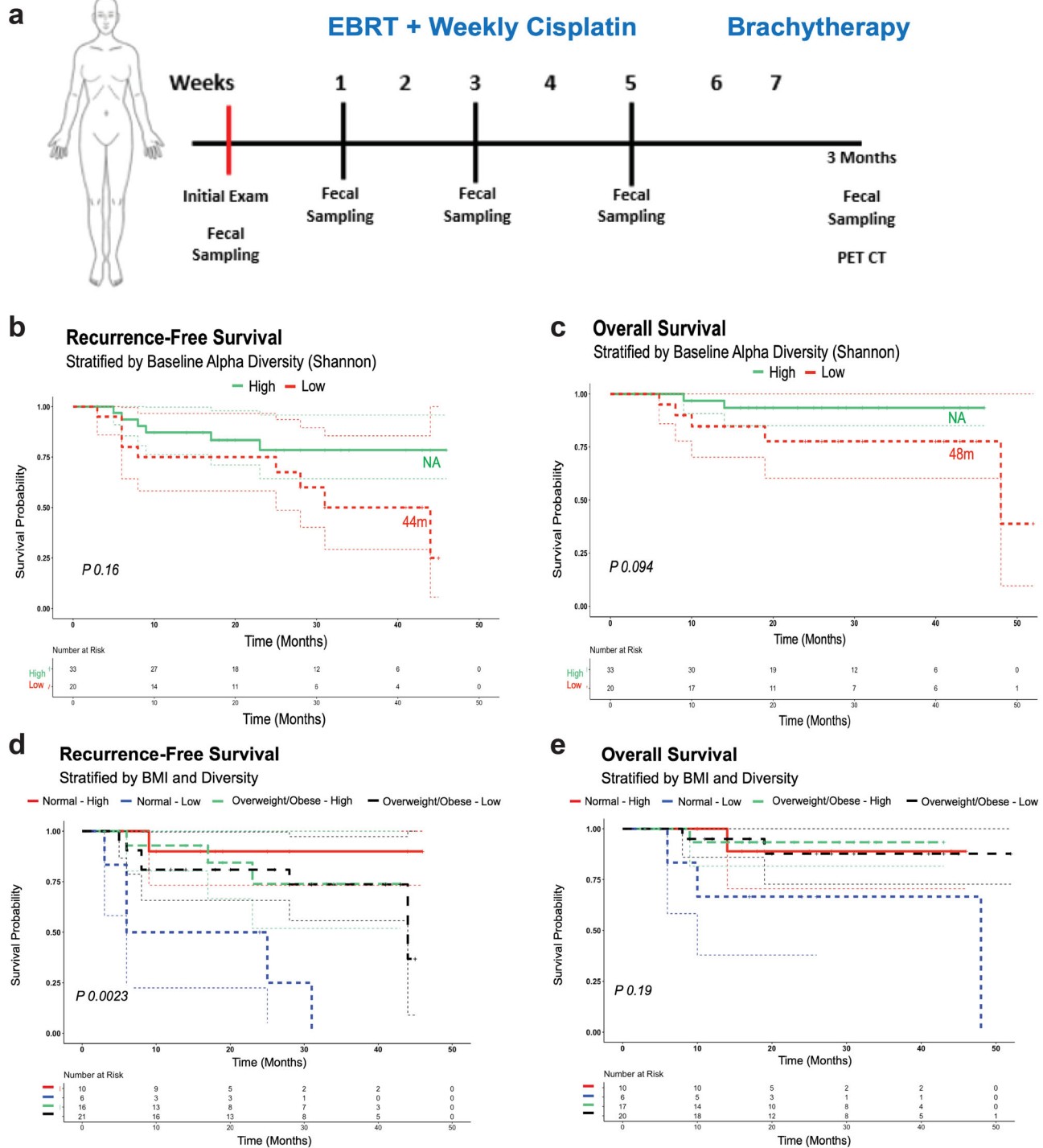

**Fig. 1 Relationship between gut diversity and BMI. a** Schema of the sample collection, treatment, and analyses used in the present study. Kaplan–Meier curves for **b** recurrence-free survival, **c** overall survival stratified by high and low gut diversity (SDI). Kaplan–Meier curves for **d** recurrence-free survival, **e** overall survival stratified by BMI and gut diversity. Cases represent patients. Only baseline samples were used for this analysis. BMI cutoffs were based on WHO standards (underweight < 18.5, normal 18.50–24.99, overweight ≥ 25, obese ≥ 30).

long-term survivors. Specifically, short-term survivor fecal samples were significantly enriched in *porphyromonas, porphyromonadaceae, and dialister*, whereas long-term survivor samples were significantly enriched in *Escherichia Shigella, Enterobacteriaceae, and Enterobacteriales* ($P < 0.05$; LDA score > 3.5, Fig. 2a, b). Our univariate analyses performed by Cox proportional hazard model identified *Pasteurellales, Haemophilus, and Veillonella* as independent predictors for RFS and OS. We tested the relationship between these taxa and RFS and OS in our cohort by stratifying patients based on their relative abundance at baseline (Supplemental Fig. 2). In patients with a low relative abundance of *Veillonella* at baseline the median OS was 48 months, and in patients with a high relative abundance of *Veillonella* at baseline the median OS was still not reached ($P = 0.054$).

**Table 1 Univariate and multivariate Cox regression analysis for recurrence-free survival.**

| Characteristics | Univariate model | | Multivariate model | |
|---|---|---|---|---|
| | HR (95% CI) | P value | HR (95% CI) | P value |
| Age | 0.93 (0.87–0.98) | 0.0096‡ | 0.93* (0.88–0.99) | 0.03‡ |
| BMI (kg/m$^2$) | 0.92 (0.84–1) | 0.096 | 0.87* (0.77–0.98) | 0.02‡ |
| Normal (18.5–24.9) | 1 (reference) | | — | — |
| Overweight (25–29.9) | 0.81(0.26–2.53) | 0.715 | — | — |
| Obese (30 or more) | 0.47(0.13–1.67) | 0.240 | — | — |
| Race/ethnicity | | | | |
| Asian | 1 (reference) | | — | — |
| Black | 0.37(0.02–5.90) | 0.479 | — | — |
| Hispanic | 0.39 (0.05–3.21) | 0.382 | — | — |
| White | 0.39 (0.05–3.31) | 0.390 | — | — |
| Other | 4.1309E-08(-inf - +inf) | 0.998 | — | — |
| Stage | | | | |
| I | 1 (reference) | | — | — |
| II | 1.50 (0.31–7.34) | 0.615 | — | — |
| III | 3.99 (0.80–20.01) | 0.091 | — | — |
| IV | 2.54 (0.23–28.12) | 0.447 | — | — |
| Grade | | | | |
| Well | 1 (reference) | | | |
| Moderate | 55297546(-inf - +inf) | 0.998 | — | — |
| Poor | 97336741.9(-inf - +inf) | 0.998 | — | — |
| Unknown | 76285161.4(-inf - +inf) | 0.998 | — | — |
| Histology | | | | |
| Squamous | 1 (reference) | | — | — |
| Adenocarcinoma/adenosquamous | 1.06(0.34–3.34) | 0.918 | — | — |
| Node level on PET | | | | |
| Common iliac | 1 (reference) | | — | — |
| External iliac | 1.33 (0.35–4.95) | 0.675 | — | — |
| Internal iliac | 0.67 (0.07–6.89) | 0.736 | — | — |
| Para-aortic | 1.31 (0.14–12.55) | 0.818 | — | — |
| None | 0.34 (0.06–2.09) | 0.245 | — | — |
| Max tumor dimension on MRI | 1.3 (1–1.8) | 0.042 | — | — |
| Smoking status | | | | |
| Current | 1 (reference) | | — | — |
| Former | 0.91 (0.10–7.84) | 0.934 | — | — |
| Never | 0.89(0.11–7.17) | 0.909 | — | — |
| Antibiotic use | | | | |
| No | 1 (reference) | | — | — |
| Yes | 78371200.7 (-inf - +inf) | 0.998 | — | — |
| Brachytherapy | | | | |
| HDR | 1 (reference) | | — | — |
| PDR | 1.41 (0.48–4.149) | 0.532 | — | — |
| Baseline gut alpha diversity | | | | |
| Observed OTU | 0.99 (0.97–1) | 0.21 | — | — |
| Shannon | 0.51 (0.23–1.1) | 0.087 | 0.36* (0.15–0.84) | 0.018‡ |
| Simpson | 0.025 (0.000036–1.7) | 0.087 | — | — |
| Inverse Simpson | 0.93 (0.84–1) | 0.11 | — | — |
| Fisher | 0.95 (0.88–1) | 0.23 | — | — |
| Camargo | 13 (0.14–1300) | 0.27 | — | — |
| Pielou | 0.02 (0.00026–1.6) | 0.081 | — | — |

*CI* Confidence interval, *HR* hazard ratio, *BMI* body mass index, *PET* positron emission tomography, *MRI* magnetic resonance imaging, *HDR* high dose rate, *PDR* pulsed dose rate, *OTU* operational taxonomic units.
*Significant hazard ratios.
‡Significant P value <0.05.

**Association between gut microbiota profile and immune signatures.** High SDI was positively correlated with tumor infiltration of CD4+ CD69+ and CD4+ PD1+ T-cells at week 3, and CD4+ ki67+ T-cells at week 5 ($P = 0.004$, $Q = 0.14$; $P = 0.036$, $Q = 0.21$; $P = 0.004$, $Q = 0.07$, respectively; Table 3). Testing for a correlation between SDI and CD4+ showed no significant associations (Fig. 3a). However, Spearman correlation showed a positive correlation between SDI and CD4+ Ki67+, CD4+ CD69+, and CD4+ PD1+ with higher Shannon diversity correlating with higher T cells (Fig. 3b–d).

## Discussion

The aim of this study was to identify independent gut microbial risk factors in CC patients receiving CRT and to evaluate their impact on survival. We found BMI and gut diversity to be independent risk factors for RFS and OS in CC patients undergoing CRT. Higher alpha gut diversity at baseline correlated with an improved RFS and OS. Our results indicate that overweight or obesity is a favorable prognostic factor independent of gut diversity. In addition, our results demonstrate that patients with better clinical survival exhibit higher diversity as well as a distinct

**Table 2 Univariate and multivariate Cox regression analysis for overall survival.**

| Characteristics | Univariate model | | Multivariate model | |
|---|---|---|---|---|
| | HR (95% CI) | P value | HR (95% CI) | P value |
| Age | 0.95 (0.87–1) | 0.23 | — | — |
| BMI (kg/m$^2$) | 0.83 (0.69–1) | 0.055 | 0.78$^*$ (0.623–0.97) | 0.025$^‡$ |
| Normal (18.5–24.9) | 1 (reference) | | — | — |
| Overweight (25–29.9) | 0.23(0.08–2.32) | 0.323 | — | — |
| Obese (30 or more) | 0.42 (0.06–4.56) | 0.19 | — | — |
| Race/ethnicity | | | | |
| Asian | 1 (reference) | | — | — |
| Black | 4.46E-09 (-inf - +inf) | 0.999 | — | — |
| Hispanic | 0.23(0.02–2.22) | 0.204 | — | — |
| White | 0.17 (0.02–1.90) | 0.151 | — | — |
| Other | 4.48E-09 (-inf - +inf) | 0.999 | — | — |
| Stage | | | | |
| I | 1 (reference) | | — | — |
| II | 1.19 (0.12–11.43) | 0.881 | — | — |
| III | 1.49 (0.09–23.93) | 0.776 | — | — |
| IV | 5.13 (0.32–82.34) | 0.248 | — | — |
| Grade | | | | |
| Well | 1 (reference) | | — | — |
| Moderate | 116103697.1 (-inf - +inf) | 0.999 | — | — |
| Poor | 46065187.92(-inf - +inf) | 0.999 | — | — |
| Unknown | 149251105.9(-inf - +inf) | 0.999 | — | — |
| Histology | | | | |
| Squamous | 1 (reference) | | — | — |
| Adenocarcinoma/adenosquamous | 3.40 (0.69–16.90) | 0.134 | — | — |
| Node level on PET | | | | |
| Common Iliac | 1 (reference) | | — | — |
| External Iliac | 1.099 (0.09–12.86) | 0.306 | — | — |
| Internal Iliac | 4.83 (0.24–98.040) | 0.999 | — | — |
| Para-aortic | 5.9333E-08 (-inf - +inf) | 0.354 | — | — |
| None | 3.34(0.26–42.69) | 0.940 | — | — |
| Max tumor dimension on MRI | 1.2 (0.77–1.8) | 1.2 | — | — |
| Smoking status | | | | |
| Current | 1 (reference) | | — | — |
| Former | 106318829.2 (-inf - +inf) | 0.999 | — | — |
| Never | 61091037.65 (-inf - +inf) | 0.999 | — | — |
| Antibiotic use | | | | |
| No | 1 (reference) | | — | — |
| Yes | 0.53 (0.06–4.56) | 0.564 | — | — |
| Brachytherapy | | | | |
| HDR | 1 (reference) | | — | — |
| PDR | 0.89 (−1.61–1.39) | 0.884 | — | — |
| Baseline gut alpha diversity | | | | |
| Observed OTU | 0.98 (0.95–1) | 0.14 | — | — |
| Shannon | 0.34 (0.1–1.1) | 0.08 | 0.19$^*$ (0.043–0.83) | 0.028$^‡$ |
| Simpson | 0.0059 (1.2e−05–2.9) | 0.1 | — | — |
| Inverse Simpson | 0.85 (0.7–1) | 0.13 | — | — |
| Fisher | 0.91 (0.79–1) | 0.15 | — | — |
| Camargo | 2200 (0.84–5800000) | 0.055 | — | — |
| Pielou | 0.0036 (5e−06–2.5) | 0.093 | — | — |

*CI* Confidence interval, *HR* hazard ratio, *BMI* body mass index, *PET* positron emission tomography, *MRI* magnetic resonance imaging, *HDR* high dose rate, *PDR* pulsed dose rate, *OTU* operational taxonomic units.
∗Significant hazard ratios.
‡Significant *P* value <0.05.

gut microbiome composition. At last, the association between gut diversity and local immune signatures suggests that patients with high gut diversity develop increased infiltration of activated CD4+ T-cell subsets and highlights helper CD4+ T cells as potential mediators of antitumor immunity upon CRT treatment. Taken together, our results imply that the diversity of gut microbiota and increased infiltration of activated CD4+ T cells might be a shared benefit factor in those who respond well to CRT treatment.

It is now generally accepted that the gut microbiome modulates immune responses, antitumor immunity, and clinical outcomes in a variety of malignancies[9,11,22]. The gut microbiome is thought to affect both innate and adaptive immune responses. Specifically, how the gut microbiome exerts its influence continues to be explored, but this explanation may have important implications if specific taxa are found to change host response to treatment via immunomodulation[6]. In our study, T helper cell profiles at baseline correlate with gut diversity. These results confer that

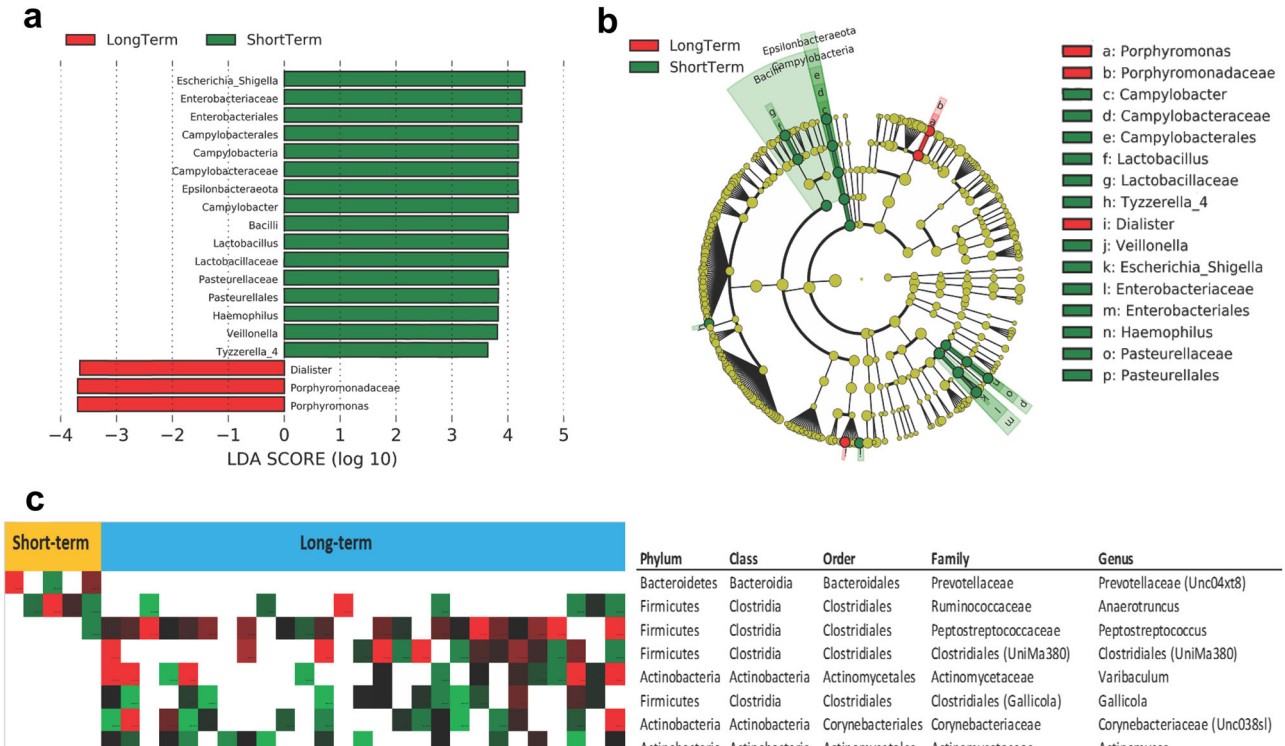

**Fig. 2 Compositional differences of the gut microbiome in short and long-term survivors. a** The different abundance of bacterial taxa between the two groups were identified by LEfSe. It was significantly different when alpha value of the factorial Kruskal–Wallis test was <0.05 and the logarithmic LDA score was >3.5. The left histogram showed the LDA scores of taxa differentially abundant between the two groups. The taxonomy was listed, followed by its core group. Putative species (Specific OTUs) identified as significantly more enriched/depleted (Fisher/Wilcoxon test *p* value < 0.05) in patients with short-term vs long-term in baseline samples. **b** Cladogram representation of the significantly different taxa features from phylum (inner circle) to genus (outer circle). **c** The right heatmap showed the relative abundance of specific bacteria by phylum, class, order, family, and genus between short-term and long-term survivors.

**Table 3 Correlation of baseline gut diversity (Inverse Shannon Diversity) with phenotype of tumor-infiltrating lymphocytes during chemoradiation treatment.**

|                   | *P* value* | *Q* value | *R*² |
|-------------------|------------|-----------|------|
| CD4+ Ki.67+ at T4 | 0.004‡     | 0.0714    | —    |
| CD4+ CD69+ at T3  | 0.004‡     | 0.1429    | —    |
| CD4+ PD1+ at T3   | 0.0367‡    | 0.2143    | —    |
| CD4+ CTLA4+ at T3 | 0.057      | 0.2857    | —    |
| CD4+              | —          | —         | 0.1  |

The percent of live lymphocytes expressing each marker was correlated with baseline Shannon diversity of the gut microbiome.
*P value correlation of immune metric with baseline gut diversity.
‡Significant P value <0.05.

T cells and response to CRT are likely affected by the gut microbiota independent of other factors such as BMI. Using multi-color flow cytometry we performed correlation analysis on individual immune signatures and microbiome diversity. The frequency of helper CD4+ T cells was chiefly identified. CC is considered to be an immunogenic tumor because its origin is dependent on a persistent infection with human papillomavirus (HPV), most often HPV16 or HPV18[23]. Previous studies have reported that the number and functional orientation of tumor-infiltrating CD4+ and CD8+ T cells and the presence of M1 type macrophages strongly correlates with survival in patients with CC after CRT[23,24]. T cells are capable of rapid antigen-specific responses and play critical roles in immune recall responses. In

addition to the percentage of CD4+ T-cell subsets, the increase in CD4 Ki67, CD4 CD69, and CD4 PD1 in patients with high microbiota diversity implies that the gut microbiome also modulates the proliferation of certain immune cell populations. Recent studies have already reported that chemoradiotherapy for CC induces unfavorable immune changes reflected by a decreased number of circulating lymphocytes, both CD4+ and CD8+ T cells, and an increased percentage in myeloid-cell populations, including myeloid-derived suppressor cells and monocytes[23]. Our study suggests that CD4+ T cells infiltrating the tumor microenvironment support and encourage the activity of other immune cells by releasing T-cell cytokines.

We found gut diversity to be associated with a favorable response to CRT against CC. Considering the correlation between gut diversity and local helper T cells being reshaped upon CRT treatment, we propose that patients harboring a more diverse gut microbiota at baseline may benefit from CRT to a greater extent. This might be mediated by the reprogramming of local antitumor immune responses. The significance of our study lies in that the modulation of gut microbiota before treatment might provide an alternative way to enhance the efficacy of CRT, specifically in advanced staged disease in which systemic failure of current therapies represents a major challenge. Our results suggest that changes in the gut microenvironment contribute substantially to treatment success or failure, particularly in so-called immunogenic tumors like CC.

Our own group has previously characterized the gut microbiome of CC patients compared with healthy female controls, and have reported on differences in the relative abundance of specific taxa[25]. Our new findings support the hypothesis that organisms

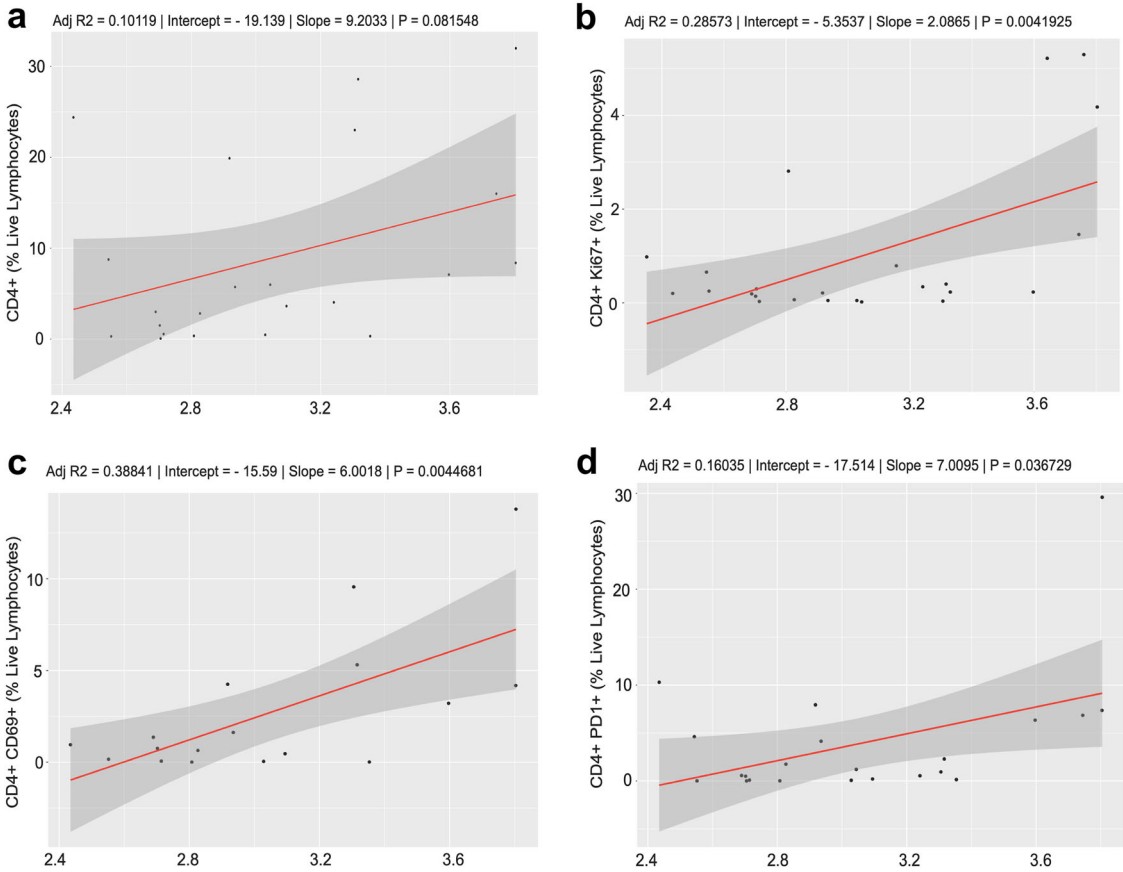

**Fig. 3 Correlation analysis of Shannon Diversity Index with tumor immune signatures. a–d** Spearman correlations between Shannon Diversity Index and CD4+, CD4+ Ki67+, CD4+ CD69+, and CD4+ PD1+. Statistical analysis was performed by Spearman correlation or Mann–Whitney tests.

inhabiting the gut microbiome may be manipulated to improve cancer treatment response. We have previously shown how CRT induces structural and compositional changes on the gut microbiome of patients with gynecologic cancers[19]. Knowing specific gut microbial organisms that inhabit and undergo changes in patients with CC during CRT provides further insight into mechanisms that may modulate immune response and potentiate treatment outcomes in cancer patients. In our analysis, long-term survivor samples were significantly enriched in *Escherichia Shigella, Enterobacteriaceae,* and *Enterobacteriales*. Previous evidence suggests distinct gut organisms can enhance systemic and antitumor immune response. This is thought to be mediated by increased antigen presentation and improved effector T-cell function in the tumor microenvironment, which may modulate treatment outcomes[11]. Researchers have already studied this treatment-enhancing utility of the gut microbiota in multiple areas of medicine[10,26]. In addition, there are emerging data describing the influence of the gut microbiome as it pertains to radiotherapy[27]. Given that radiation can change the composition of the gut microbiome by altering the relative abundance of different taxa, we have to postulate whether radioresistant taxa ultimately alter the effectiveness of radiotherapy for CC[6,28,29]. The results of our study illustrate the potential of intentionally modifying the gut microbiota to accumulate CRT-tolerant species as an interventional strategy to enhance response of CC to CRT. Furthermore, determining whether changes in the human gut microbiome during CRT affect patients' risk of treatment-related toxic effects may be an area that deserves further investigation. Recently, our collaborative group investigated the association between patient-reported bowel toxicity and the gut microbiota at baseline in thirty-five patients receiving radiation therapy for the

definitive management of CC[30]. We found increased radiation toxicity to be associated with a decrease in gut microbiome diversity. In addition, although baseline gut diversity was not predictive of end-of-treatment bowel toxicity, we identified compositional characteristics associated with members of the *Clostridiales* order in patients who experienced less toxicity. Finding ways to combat radiation-associated toxicity is of great interest to overall patient well-being. For radiation, GI toxicity that is not amendable to standard treatment, modulation of gut microbiota before CRT might provide an alternative way to enhance not only treatment efficacy but improve treatment-associated toxicity in CC patients.

The "obesity paradox", which suggest a positive association between increasing BMI as it pertains to a specific disease, was firstly reported in heart failure[31], but has since been described in a variety of disease processes, including other gynecologic cancers[32–34]. Theories centered around the "obesity paradox" suggest that patients with a high BMI may be better able to withstand cancer-induced consumption and stress compared with patients with a low BMI[35]. In uterine cancer, it has been reported that the risk of recurrence differed significantly by BMI[36]. Specifically, a greater proportion of obese women met the criteria for having a low risk of recurrence, while thin women tended to have a high-intermediate risk or recurrence. Many studies have investigated the impact of BMI on CC, but the association between weight and CC remains ambigous[37]. HPV is considered to be responsible for 99.7% of all CCs[38], however, it has been suggested that obesity may further increase this risk[39–41]. Other reports however do not report an association[42,43]. For example, a review by Lane et al.[44] finally refuted the relationship between CC and obesity ultimately citing a lack of evidence. The inconsistent conclusions among

studies investigating the association between BMI and CC may be attributed to numerous factors including patient selection criteria, sample size, and generalizability of the study population. Among these factors, patient selection criteria may be especially important, because tumor histology seems to be closely associated with BMI[41].

We found age to be an independent risk factor for RFS in CC patients undergoing CRT. We previously characterized the fecal microbiome in women with CC and found the diversity of the fecal microbiome to differ between young and older women[45]. Others have found advanced age to be associated with variations in the gut microbiota composition characterized by a loss of diversity of specific taxa[46]. Our finding of age as an independent risk factor for RFS in CC may signify a temporal association between an altered bacterial composition in the gut and response to therapy.

The strengths of this study include the use of careful clinical staging, histopathology, and reliable phylogenetic and statistical analysis to assess bacterial community compositional changes using both microbial divergence and taxon-based methods. In addition, we followed a complete protocol for 16S analysis ranging from the sample collection method, DNA extraction, and microbiome sequencing, thus limiting artifactual variations. Our group previously compared data from 16 s rRNA and whole-genome sequencing (WGS)[47]. We found that diversity, evenness, and richness measures are comparable when using 16 s rRNA and WGS, whereas relative abundances of individual genera differed between the two methods. Based on the comparisons, we found that 16 s rRNA is more accurate in taxonomic classifications when looking at proportions and abundances. Although this study has yielded intriguing findings, an important limitation is the small sample size. Consequently, the sample size limited our ability to weigh statistical power. In addition, details such as place of residence, lifestyle, and diet were not known in our data set. Furthermore, we acknowledged the observation that factors such as racial variation have been shown to contribute to differences in host genetics and innate/adaptive immunity and could have impacted our overall outcomes. Yet despite these limitations and the relatively small size of this prospective study, large statistically significant differences were still observed, and we believe the results presented herein provide solid evidence elucidating the role of the gut microbiome as it pertains to the treatment of CC. We hope the integration of these data will produce actionable strategies geared toward targeting and manipulating the microbiome in order to ultimately improve CC therapy.

In conclusion, our study demonstrates that the diversity of gut microbiota is associated with a favorable response to CRT. In addition, compositional variation among patients correlated with short-term and long-term survival. Our study demonstrates that gut diversity is a significant factor for predicting OS in CC patients undergoing CRT when BMI is accounted for and may help explain the "obesity paradox" in cancer response. Moreover, analysis of immune cells from cervical tumor brush samples by flow cytometry revealed an association between high microbial diversity, increased tumor infiltration of CD4+ lymphocytes and the activation of CD4 cells over the course of radiation therapy. The correlation between gut diversity and increased tumor infiltration of CD4+ lymphocytes suggest that patients harboring a more diverse gut microbiota at baseline may benefit from CRT to a greater extent. The significance of our study lies in that modulation of gut microbiota before CRT might provide an alternative way to enhance treatment efficacy and improve treatment outcomes in CC patients. Additional studies exploring the relationship between gut diversity, CRT, and treatment efficacy are needed to further understand the role of the gut microbiome in CC treatment.

## Methods

**Participants and clinical data.** Gut microbiome and cervical swab samples were collected prospectively from CC patients according to a protocol approved by The University of Texas MD Anderson Cancer Center Institutional Review Board (MDACC 2014-0543) for patients with biopsy-proven carcinoma of the cervix treated at MD Anderson and the Lyndon B. Johnson Hospital Oncology Clinic from 22 September 2015 to 11 January 2019. Written informed consent from eligible patients willing to participate in this study was obtained. Inclusion criteria included all patients who had new diagnoses of locally advanced, nonmetastatic carcinoma of the cervix and underwent definitive CRT with EBRT followed by brachytherapy. None of the patients included in this cohort underwent surgery. Ineligibility criteria included currently pregnant or lactating women. Patients received a minimum of 45 Gy via EBRT in 25 fractions over 5 weeks with weekly cisplatin followed by two brachytherapy sessions at approximately weeks 5 and 7 with EBRT in between for gross nodal disease or persistent disease in the parametrium. Patients with stage IB1 cancer were given CRT due to the presence of nodal disease. Medical history and current medication use were assessed via an in-person interview with a clinical provider or trained study staff. Clinical variables, demographics, and pathologic reports were abstracted from electronic medical records.

**Sample collection and DNA extraction.** Rectal swabs were collected from all patients by a clinician performing rectal exams at five time points (baseline; weeks 1, 3, and 5 of radiotherapy; and 3 months after CRT completion) using a matrix-designed quick-release Isohelix swab to characterize the diversity and composition of the microbiome over time (product # DSK-50 and XME-50, www.isohelix.com, UKSamples). The swabs were stored in 20 μl of protease K and 400 μl of lysis buffer (Isohelix) and kept at −80 °C within 1 h of sample collection.

**16 S rRNA gene sequencing and sequence data processing.** 16 S rRNA sequencing was performed for fecal samples obtained from all patients at four time points to characterize the diversity and composition of the microbiome over time. 16 S rRNA gene sequencing was done at the Alkek Center for Metagenomics and Microbiome Research at Baylor College of Medicine. 16 S rRNA was sequenced using approaches adapted from those used for the Human Microbiome Project[4]. The 16 S rRNA V4 region was amplified via polymerase chain reaction with primers that contained sequencing adapters and single-end barcodes, allowing for pooling and direct sequencing of polymerase chain reaction products. Amplicons were sequenced on the MiSeq platform (Illumina) using the 2 × 250-bp paired-end protocol, yielding paired-end reads that overlapped nearly completely. Sequence reads were demultiplexed, quality-filtered, and subsequently merged using the USEARCH sequence analysis tool (version 7.0.1090) (4). 16 S rRNA gene sequences were bundled into operational taxonomic units at a similarity cutoff value of 97% using the UPARSE algorithm[48]. 16 S amplicon sequences were processed using the Center for Metagenomic and Microbiome Research's 16 S analysis pipeline which is based on the UPARSE[49,50] clustering algorithm. In brief, amplicon reads are trimmed at $Q = 5$ and merged allowing three mismatches in the overlapping region. The resulting merged reads are filtered against an expected error filter of 0.05 across the entire merged read. Only merged reads of length between 252 and 254 bps are kept for downstream processes. The resulting readset is dereplicated and iteratively clustered using UPARSE in one bp step up to an 8 bp radius (3.2% radius). Counts per OTU are tabulated per sample and the resulting data set is processed with ATIMA2 (Agile Toolkit for Incisive Microbiome Analyses (https://github.com/cmmr/atima), which provides an integrated solution for exploring relationships between microbial communities and emergent properties of their hosts or environments. All the microbiome metrics presented in this manuscript were calculated using ATIMA2 except for the community evenness metrics Camargo and Pielou[51,52] which were calculated using the function "evenness()" in the R package "microbiome". To generate taxonomies, operational taxonomic units were mapped to an enhanced version of the SILVA rRNA database containing the 16Sv4 region. A custom script was used to create an operational taxonomic unit table from the output files generated as described above for downstream analyses of α-diversity, β-diversity, and phylogenetic trends. Principal coordinates analysis was performed by institution and sample set to make certain no batch effects were present.

**Flow cytometry.** In order to explore the association between the gut microbiota and immune signatures, we analyzed the cervical tumors in our cohort of patients via flow cytometry on tumor brushings performed before week 1, week 3, and week 5 of radiation therapy. Immunostaining was performed according to standard protocols[53]. Cells were fixed using the Foxp3/Transcription Factor Staining Buffer Set (eBioscience, Waltham, MA) and stained with a 16 color panel with antibodies from Biolegend (San Diego, CA), BD Bioscience (San Jose, CA), eBioscience (Waltham, MA), and Life Technologies (Carlsbad, CA). Analysis was performed on a 5-laser, 18 color LSRFortessa X-20 Flow Cytometer (BD Biosciences, San Jose, CA). Analysis was performed using FlowJo version 10 (Flowjo LLC, Ashland, OR). We then followed a similar previously published method[53]. In brief, the cells were incubated with the antibodies for surface markers at 4 °C in dark for 30 minutes. They were then washed twice with fluorescence-activated cell sorting (FACS) buffer

and fixed and permeabilized with FOXP3 Fix/perm Kit (ThermoFisher Scientific, Waltham, MA). Next, intracellular staining was performed by preparing the antibodies in permeabilization buffer and incubating the cells for 30 minutes at 4 °C in the dark. Cells were washed with FACS buffer twice and prepared for acquisition on an LSR Fortessa X-20 analyzer at the Flow Cytometry Core at MD Anderson Cancer Center and were analyzed using FlowJo software (FlowJo, LLC, Ashland, OR). Compensation controls were prepared using OneComp ebeads (eBioscience, Waltham, MA) and fluorescence minus one controls were used[53].

**Statistics and reproducibility**. For microbiome analysis, rarefaction depth was set at 7066 reads. The SDI was used to evaluate α-diversity (within samples), and principle coordinates analysis of unweighted UniFrac distances was used to examine β-diversity (between samples). A ROC curve was calculated using a linear regression model. A table was constructed using the coordinates of the curve in order to identify the most appropriate cutoff for SDI. The cutoff for SDI was identified to be 2.69. Patient and tumor characteristics were analyzed by univariate and multivariate Cox regression models for RFS and OS. Covariates with a $p$ value of ≤ 0.1 in the univariate model were chosen to be included in the multivariate analysis. RFS was defined as the time from the initiation of the therapy until the date of disease relapse or progression or death from any cause. Disease relapse or progression was defined as appearance or signs of the disease that were confirmed pathologically or radiographically. OS is defined as the time from the date of initiation of therapy until the date of death from any cause. Survival was estimated by the Kaplan–Meier method, and comparisons were made by log-rank tests. Characteristics included age, BMI, race, FIGO stage, grade, histology, nodal status, smoking status, antibiotic use, and max tumor size. For each outcome of interest, a multivariate Cox regression analysis was performed to adjust for the effects of prognostic factors identified on univariate analysis as influencing survival in CC. These analyses were conducted using covariates with $p ≤ 0.1$ in a stepwise fashion. We also ran a correlation analysis of alpha diversity metrics with tumor flow cytometry markers using a linear regression and Spearman's correlation. Alpha (within sample) diversity was evaluated using SDI. The relative abundance of microbial taxa, classes, and genera present in long-term vs short-term survivors was determined using LDA Effect Size[54], applying the one-against-all strategy with a threshold of 3.5 for the logarithmic LDA score for discriminative features and α of 0.05 for factorial Kruskal–Wallis testing among classes. Long-term survivors were classified as patients who had a follow-up of 2 years or more and were alive at time of last follow-up, whereas short-term survivors had a follow-up of one year or less. LDA Effect Size analysis was restricted to bacteria present in 20% or more of the study population. Kaplan–Meier curves were generated for patients with normal BMI and overweight/obese BMI based on Cox analysis and clostridia abundance (BMI cutoffs were based on WHO standards). The significance of differences was determined using the log-rank test. Statistical significance was set at an alpha of 5% for a two-sided $p$ value. Analyses were conducted using Rstudio version Orange Blossom—1.2.5033.

**Reporting summary**. Further information on research design is available in the Nature Research Reporting Summary linked to this article.

## Data availability

The sequencing data sets generated during and/or analyzed during the current study are available in the Sequence Read Archive repository (accession number: PRJNA685389). All other data generated or analyzed during this study are included in this published article (and its supplementary information files).

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

## Acknowledgements

This research was supported in part by the Radiological Society of North America Resident/Fellow Award (to L.E.C.), the National Institutes of Health (NIH) through MD Anderson's Cancer Center Support Grant P30CA016672, the Emerson Collective and the National Institutes of Health T32 grant #5T32 CA101642-14 (T.T.S.). This study was partially funded by The University of Texas MD Anderson Cancer Center HPV-related Cancers Moonshot (L.E.C. and A.H.K.). The human subjects who participated in this study are gratefully acknowledged.

## Author contributions

A.Y.D.M., T.S., and M.A.K. were responsible for subject identification and sample collection. S.D.E., V.L.H., S.N., K.Y.C., X.W., B.V.C., K.J.S., G.W.G.B., A.Y.D.M., T.S., and M.A.K. carried out laboratory testing. T.V.K., X.W., N.J.A., L.E.C., A.H.K., M.B.E.A., and T.T.S. analyzed the data. T.V.K., S.D.E., V.L.H., S.N., K.Y.C., X.W., B.V.C., K.J.S., G.W.G.B., A.Y.D.M., T.S., M.P.M., J.F.P., L.L.L., L.R., A.J., K.M.S., N.J.A., J.W, L.E.C., A.H.K., M.B.E.A., and T.T.S. were responsible for the interpretation of the statistical analysis, and review and approval of the final manuscript. The study concept was conceived by L.E.C., A.H.K., M.B.E.A., and T.T.S. The manuscript was written by T.T.S.

## Competing interests

The authors declare no competing interests.
