## [Peer Review File · Communications Biology]

Reviewers' comments:

Reviewer #1 (Remarks to the Author):

Thanks for letting me review this article. It is an interesting study to understand associations between the gut microbiome and cancer survival outcomes. I have some major comments as below:

Introduction:

1) More information is needed for the mechanisms of the gut microbiome associated with cancer RT. As CRT was used in this study, there is no information how chemo associated with gut microbiome.

2) It is not clear how the gut microbiome diversity and abundance changes with CRT, please provide more information. As a variety of factors can influence the gut microbiome, there is no mention these factors and also what type of factors will be involved in this study.

Methods:

1) The inclusion and exclusion criteria of women are not clear. Did they receive surgeries as well? For these with antibiotics, the gut microbiome will be totally impacted.

2) When the recurrence free survival and overall survival were collected? As a gut microbiome study, it is very big limitation without mentioning any information about diet.

3) As the gut microbiome data were collected longitudinally, no information was provided how the analysis was done and how they correlated with survival and inflammation factors?

4) For the cox analysis, some of these variables with such a small sample size, for example Asian only has 2 subjects. How these small numbers influence the results?

5) For the microbiome analysis, Qiime2 has been recommended in current studies. The authors seem to use Qiime 1 in this study. An updated version of analysis may be appropriate for this study.

Results:

1) A very big problem is the longitudinal data of the gut microbiome. However, only baseline data were mentioned. Please revise the aims of this study if only baseline gut microbiome data were used in this study.

2) For the diversity analysis, why only use the SDI and unweighted UniFrac? Some other diversity indices should be used. Based on the models in Tables 1 and 2, evenness seems associated with the survival outcomes. Why the evenness measure was not selected in the cox survival analysis?

3) Another limitation of this study is 16S rRNA, which cannot provide strain or species level data.

4) This study also has a problem with respect to participant distributions. For example, supplemental Figure 1, you can find out such a small number of participants in histology, grade, and stage.

5) In this study, many of the results showed trends of significance. The results should be explained very carefully.

Discussion:

1) Gut microbiome is associated with obesity based on previous literature. When explaining the obesity paradox, it brings out another question whether this gut microbiome difference was caused by obesity. Also, there is a conflict in relationships between BMI (continuous vs categorical variables) and survival outcomes, please explain this.

2) Results seem to miss the associations between the gut microbiome + immune cells and survival outcomes.

Reviewer #2 (Remarks to the Author):

In this manuscript entitled "Gut microbiome diversity is an independent predictor of survival in cervical cancer patients receiving chemoradiation" by Dr. Travis T. Sims et al., the authors explored the associations between gut microbiome and clinical outcomes in patients with cervical cancer receiving chemoradiotherapy. They found that the diversity of gut microbiome (in Shannon diversity index) as a prognostic factor for both recurrence and overall survival. As far as my knowledge, there is a paucity of data on the prognostic impact of gut microbiome in cancer patients receiving chemoradiotherapy (without immunotherapy). The manuscript contains

interesting findings and the data would provide some insight in this field. However, in my opinion, substantial revisions are needed to consider this manuscript for publication. Please see my comments below. I hope my comments are useful for improvement of the article.

Major comments

1. The authors concluded that gut microbiome diversity is an independent predictor of survival in cervical cancer patients receiving chemoradiation, however this conclusion might have altered depending on the predefined cutoff points of p-value for the survival analysis. The authors seem to have defined significant p-value as <0.1 . I understood the setting looser (than <0.05) p-value can be an option considering the relatively small number of patients in total, but this part needs to be rationalized to lead to firm conclusions as above. At least, the authors should add careful explanations on the limitations of the study. Also, the author may be encouraged to soften the expression for the prognostic impact of microbiome in discussion section as well as the main title.

2. The high percentage of antibiotic use could have an impact on the results from this study. The authors should mention this in the discussion and provide the details (types etc.) of the antibiotics.

Minor comments

1. Page 6, line 116: There is a discordance between the cutoff of p-value of univariate analysis in result ($P \leq 0.2$) and Method (<0.1). This would be critical and the authors should clarify which is the predefined p-value.

2. Page 6, line 132-134: The authors need to describe the methods of calculation by receiver operating characteristic curve. Also, the authors are encouraged to provide the ROC as a supplemental material.

3. Page 6, line 134-142: The authors should describe the survival differences referring to their statistical differences (just numerically or statistically) according to their predefined level, not just as "longer".

4. Page 7, line 155-157: The authors should describe the survival differences referring to their statistical differences (just numerically or statistically) according to their predefined level, not just as "prolonged".

5. Page 7, line 159-160: The following part should not be included in the result section: "Because the gut microbiota is thought to influence disease progression partially through modulating local immune response". This information should be given in the intro section with appropriate references.

6. Page 7, line 160- page 8, line 163: The following part should not be included in the result section: "we analyzed the cervical tumors in our cohort of patients via flow cytometry on tumor brushings performed before week 1, week 3 and week 5 of radiation therapy. To identify features associated with high gut diversity, Spearman correlation analysis was conducted between immune signatures at each time point". This information should be given in the method section.

7. Page 8, line 165-166: The following part should not be included in the result section: "These results suggest that patients with high gut diversity develop increased infiltration of activated CD4+ T-cell subsets". This speculation should be provided in the discussion.

8. Page 13, line 13: Please provide the company and catalog number for the swab and lysis buffer the authors used in this study.

9. Figure 1: The authors are encouraged to add the number at risk below for each K-M curve. This would help the readers to easily grasp the sample size in each subgroup and more appropriate interpretation of the data. The cutoff of BMI should be stated in figure legends (or in figures). "Diversity" should be replaced to "Baseline alpha-diversity".

10. Supplemental Fig 1: Diversity -> Alpha diversity

11. Table1, 2: In the row of characteristics, please provide the number of patients categorized into each subgroup. Also, abbreviation should be provided (not only for CI and HR). For "significant P value", the author should specify the cutoff value in the footnote of the table

12. Supplemental table 1: The percentages for the patients seem to be wrong in the subgroup of Race (Asian, Black). Please double check for the other subgroups.

Reviewer #3 (Remarks to the Author):

Thank you for the opportunity to review this manuscript. The authors have provided a clear and concise dataset further implicating the microbiota in treatment outcomes for cancer. They have specifically built upon existing research to demonstrate a specific microbial contribution to radiotherapy efficacy, an approach that to date has not been explored well. I believe these findings will be relevant to the broader research community, prompting new approaches aimed at modulating the microbiota to improve outcome.

The authors have provided a convincing dataset despite the fairly low sample size, and their approach is complemented by immune analysis to provide mechanistic insight regarding how the microbiota confers its effect on treatment efficacy.

What is disappointing about this paper is the siloed approach to outcome, with survival discussed without any mention of toxicity. The microbiota is well documented to contribute to acute and chronic toxicities, particularly radiation-induced gastrointestinal mucositis/proctitis. The indirectly affect survival by influencing treatment compliance, dosing schedules, etc. Evidence in women with cervical cancer already shows that women who develop toxicity have vastly different microbial communities compared to those that do not develop toxicity.

This has not been discussed in the paper at all and represents a key oversight in the approach used by authors, particularly given the overlap in toxicity and efficacy. I would recommend inclusion of toxicity data if it has been collected to determine if there is any influence of toxicity profiles on OS/PFS or if the same microbial findings can be applied to toxicity prediction. Alternatively, if this is not possible, the authors must address this in the discussion with clear reference to how their results can be applied to improve efficacy without inadvertently promoting toxicity.

Furthermore, the authors do not discuss the specifics of their microbial findings. It seems counterintuitive that "pathogenic" strains like Shigella and Enterobacteriaceae would correlate with optimal survival - could the authors please expand on this finding and postulate why/how pathogenic bacteria may promote radioefficacy. Again, considering potential immunosuppression in cancer patients, increasing the abundance of these microbes to boost efficacy may be undermined by heightened risk of infection.

Overall, a solid study with some interesting findings likely to be relevant across many areas of cancer. I wish the authors all the best.

Kind Regards
Dr Hannah Wardill

Manuscript ID: COMMSBIO-20-1670-T

Title: Gut microbiome diversity is an independent predictor of survival in cervical cancer patients receiving chemoradiation

Dear Communications Biology Reviewers,

We are very excited to have been given the opportunity to revise our manuscript entitled “Gut microbiome diversity is an independent predictor of survival in cervical cancer patients receiving chemoradiation” for Communications Biology. We want to extend our appreciation for taking the time and effort necessary to provide such insightful guidance. Herein, we explain how we revised the paper based on your comments and recommendations.

Note: Page and line numbers refer to the edited manuscript with revisions

REVIEWER 1, COMMENT 1 (Introduction)

“More information is needed for the mechanisms of the gut microbiome associated with cancer RT. As CRT was used in this study, there is no information how chemo associated with gut microbiome”

Response:

-We appreciate the reviewer’s feedback and agree that additional mechanistic insights are needed. Few studies have investigated the relationship between chemoradiation and the gut microbiome but limited existing research, including our own, support that chemoradiotherapy triggers a treatment-induced shift of the intestinal microbiota in cancer patients. The mechanism by which this shift occurs is beyond the current study, but could be related to changes in the gut epithelium and mucus layer, followed by overgrowth of pathogenic bacteria, which could in turn affect

immune cell maturation and response to tumor. Cumulative fractions of CRT may induce outgrowth of these radioresistant or pathologic microbial taxa, selecting taxa tolerant to radiation-induced insults to persist. We also agree that the relative effects of chemotherapy versus radiation therapy is of great interest. However we are unable to tease this apart as the combination therapy is the standard approach for treatment. Future pre-clinical studies may be able to address this

-Edits to the manuscript addressing these concerns can be found on page 4-5, lines 89-97.

Edits read: “Limited studies have examined the association between cancer treatment and the gut microbiome in patients with gynecological malignancies. CRT has been shown to alter overall richness and diversity of the gut microbiome in cancer patients¹⁹⁻²¹. The mechanism by which this occurs has not been fully elucidated, but cumulative fractions of CRT may induce outgrowth of radioresistant microbial taxa, ultimately selecting bacteria tolerant to radiation-induced insults to persist.”

REVIEWER 1, COMMENT 2 (Introduction)

“It is not clear how the gut microbiome diversity and abundance changes with CRT, please provide more information. As a variety of factors can influence the gut microbiome, there is no mention these factors and also what type of factors will be involved in this study.”

Response:

-We appreciate the reviewer’s feedback. Limited existing research has shown chemoradiotherapy to diminish microbiome diversity and abundance in cervical cancer patients. Four cohort trials, including our own that included patients with cervical cancer have shown a decrease in overall richness and diversity of the gut microbiome during CRT.

With respect to factors that influence the gut microbiome, age, BMI, race/ethnicity, smoking status and antibiotic use are all factors thought to influence the gut microbiome which we account for in our study. Other factors such as place of residence and lifestyle have also been shown to affect the cervical microbiome. We recognize that this is a limitation of the study and a potential source of confounding bias. As such, we acknowledged the observation that factors such as place of residence and lifestyle could impact our overall outcomes. This limitation is unlikely to fully explain the large differences in survival we observed. We have included a comment in regards to this in the limitations paragraph of the text.

-Edits to the manuscript can be found on page 10, lines 221-223 and on page 12, lines 277-278.

Edits read: “We have previously shown how CRT induces structural and compositional changes on the gut microbiome of patients with gynecologic cancers” And “Additionally, details such as place of residence, lifestyle and diet were not known in our data set. Furthermore, we acknowledged the observation that factors such as racial variation have been shown to contribute to differences in host genetics and innate/adaptive immunity and could have impacted our overall outcomes”

REVIEWER 1, COMMENT 1 (Methods)

“The inclusion and exclusion criteria of women are not clear. Did they receive surgeries as well?

For these with antibiotics, the gut microbiome will be totally impacted.”

Response:

-We appreciate the reviewer’s feedback. Inclusion criteria included all patients who had a new diagnoses of locally advanced, nonmetastatic carcinoma of the cervix and underwent definitive CRT with EBRT followed by brachytherapy. None of the patients included in this cohort underwent surgery. Ineligibility criteria included currently pregnant or lactating women. Medical

history and current medication use were assessed via an in-person interview with a clinical provider or trained study staff.

-Edits to the manuscript can be found on page 14, lines 307-311 and page 14, lines 314-316.

Edits read: “Inclusion criteria included all patients who had new a diagnoses of locally advanced, nonmetastatic carcinoma of the cervix and underwent definitive CRT with EBRT followed by brachytherapy. None of the patients included in this cohort underwent surgery. Ineligibility criteria included currently pregnant or lactating women.” And “Medical history and current medication use were assessed via an in-person interview with a clinical provider or trained study staff.”

REVIEWER 1, COMMENT 2 (Methods)

“When the recurrence free survival and overall survival were collected? As a gut microbiome study, it is very big limitation without mentioning any information about diet.”

Response:

-We appreciate the reviewer’s feedback. Recurrence free survival (RFS) was defined as the time from the initiation of the therapy until the date of disease relapse or progression or death from any cause. Disease relapse or progression was defined as appearance or signs of the disease that was confirmed pathologically or radiographically. Overall survival (OS) is defined as the time from the date of initiation of therapy until the date of death from any cause. Survival was estimated by the Kaplan–Meier method, and comparisons were made by log-rank tests. With respect to diet, we appreciate the reviewer’s feedback. We recognize that this is a limitation of the study. Unfortunately diet was not the initial focus of this study and we do not have complete baseline diet data to expand our current analysis. However, we agree with the reviewer’s critique and documentation of diet would be of interest in future validation studies. We have included a comment in regards to this in the limitations paragraph of the text.

-Edits to the manuscript can be found on page 12, lines 277-281 and page 17, lines 383-390.

Edits read: “Additionally, details such as place of residence, lifestyle and diet were not known in our data set. Furthermore, we acknowledged the observation that factors such as racial variation have been shown to contribute to differences in host genetics and innate/adaptive immunity and could have impacted our overall outcomes.” And “Covariates with a p-value of less than or equal to 0.1 in the univariate model were chosen to be included in the multivariate analysis. Recurrence free survival (RFS) was defined as the time from the initiation of the therapy until the date of disease relapse or progression or death from any cause. Disease relapse or progression was defined as appearance or signs of the disease that was confirmed pathologically or radiographically. Overall survival (OS) is defined as the time from the date of initiation of therapy until the date of death from any cause. Survival was estimated by the Kaplan–Meier method, and comparisons were made by log-rank tests.”

REVIEWER 1, COMMENT 3 (Methods)

“As the gut microbiome data were collected longitudinally, no information was provided how the analysis was done and how they correlated with survival and inflammation factors?”

Response:

-We appreciate the reviewer’s feedback. Collection details and microbiome analysis details can be found in the online methods section of the manuscript. Stool was collected from all patients by a clinician performing rectal exams at five time points (baseline; weeks 1, 3, and 5 of radiotherapy; and 3 months after CRT completion) using a matrix-designed quick-release Isohelix swab to characterize the diversity and composition of the microbiome over time. 16S rRNA sequencing was performed for fecal samples obtained from all patients at four time points to characterize the diversity and composition of the microbiome over time. 16S rRNA gene sequencing was done at

the Alkek Center for Metagenomics and Microbiome Research at Baylor College of Medicine. 16S rRNA was sequenced using approaches adapted from those used for the Human Microbiome Project. Further description of collection details and microbiome analysis including flow cytometry details can be found in the online methods section of the manuscript under “Sample collection and DNA extraction” and “16S rRNA gene sequencing and sequence data processing”.

REVIEWER 1, COMMENT 4 (Methods)

“For the cox analysis, some of these variables with such a small sample size, for example Asian only has 2 subjects. How these small numbers influence the results?”

Response:

-We recognize that this is a limitation of the study as we are unable to account for the possibility that confounding variables may account for the higher rates of survival in patient with high gut diversity. However, we accounted for as many potential variables as possible in the cox analysis and found gut diversity to be an independent predictor of survival. As such, we have included additional discussion in the paper acknowledging that factors such as racial variation may have contributed to differences in host genetics and innate/adaptive. Despite this, we believe this limitation is unlikely to fully explain the large differences in survival we observed. We appreciate the reviewer’s feedback and have included a comment in regards to this in the limitations paragraph of the text.

-Edits to the manuscript can be found on page 12, lines 278-281.

Edits read: “Additionally, details such as place of residence, lifestyle and diet were not known in our data set. Furthermore, we acknowledged the observation that factors such as racial variation have been shown to contribute to differences in host genetics and innate/adaptive immunity and could have impacted our overall outcomes.”

REVIEWER 1, COMMENT 5 (Methods)

“For the microbiome analysis, Qiime2 has been recommended in current studies. The authors seem to use Qiime 1 in this study. An updated version of analysis may be appropriate for this study.”

Response:

We thank the reviewer for the suggestion of employing alternative methods to process the 16S data. We recognize that multiple processing and analytic pipelines are available, nonetheless our close collaborators at the Center for Metagenomic and Microbiome Research have developed, validated, and implemented multiple bioinformatic tools to profile microbiomes. The methods developed and implemented by our collaborators at the CMMR have been validated against multiple mock communities and utilized to evaluate numerous datasets previously. Compared to other publicly available 16S analytic pipelines, including QIIME and QIIME2, when ran with default settings, the CMMR 16S pipeline employs stricter filtering parameters that result in a significant reduction in sequencing noise which is reflected on the microbiome metrics calculated and presented in this manuscript.

-Edits to the manuscript can be found on page 15, lines 338-350.

Edits read: “16S amplicon sequences were processed using the Center for Metagenomic and Microbiome Research’s 16S analysis pipeline which is based on the UPARSE clustering algorithm. Briefly, amplicon reads are trimmed at Q=5 and merged allowing 3 mismatches in the overlapping region. The resulting merged reads are filtered against an expected error filter of 0.05 across the entire merged read. Only merged reads of length between 252-254 bps are kept for downstream processes. The resulting readset is dereplicated and iteratively clustered using UPARSE in one bp step up to an 8 bp radius (3.2% radius). Counts per OTU are tabulated per

sample and the resulting dataset is processed with ATIMA2 (Agile Toolkit for Incisive Microbiome Analyses (<https://github.com/cmmr/atima>) which provides an integrated solution for exploring relationships between microbial communities and emergent properties of their hosts or environments. All the microbiome metrics presented in this manuscript were calculated using ATIMA2 except for the community evenness metrics Camargo and Pielou which were calculated using the function “evenness()” in the R package ‘microbiome’.”

REVIEWER 1, COMMENT 1 (Results)

“A very big problem is the longitudinal data of the gut microbiome. However, only baseline data were mentioned. Please revise the aims of this study if only baseline gut microbiome data were used in this study.”

Response:

-We appreciate the reviewer’s feedback. Reporting on the longitudinal data was not the initial focus of this study as we sought to determine if baseline diversity and composition of the gut microbiome was independently associated with survival in cervical cancer patients receiving chemoradiation. However, we agree with the reviewer’s critique and longitudinal analysis is included in the supplemental data. We hope to use this longitudinal data to expand our current analysis in future validation studies.

REVIEWER 1, COMMENT 2 (Results)

“For the diversity analysis, why only use the SDI and unweighted UniFrac? Some other diversity indices should be used. Based on the models in Tables 1 and 2, evenness seems associated with the survival outcomes. Why the evenness measure was not selected in the cox survival analysis?”

Response:

-We appreciate the reviewer's feedback. Observed OTU, Shannon diversity index (SDI), Simpson, Inverse Simpson diversity metric (ISD) and Fisher, were all used to evaluate alpha diversity with similar findings, although SDI is presented in this manuscript for simplicity (complete diversity data is included in supplemental data). As for the evenness metrics, we acknowledge that Pielou satisfied our cut-off to be included in the multivariate survival analysis, however we identified an issue of co-linearity and thus decided to not include it. We have plotted Kaplan Meier curves for Pielou and included them in supplemental.

REVIEWER 1, COMMENT 3 (Results)

"Another limitation of this study is 16S rRNA, which cannot provide strain or species level data."

Response:

-We appreciate the reviewer's comments. Previously published work has shown a sizable amount of agreement between 16S and WMS sequencing techniques and it is currently thought that 16S and WMS have a significant degree of correlation. The vast majority of microbiome analyses have utilized 16S rRNA gene sequencing (16S) which uses variable regions of the 16S ribosomal RNA gene to assign taxonomic classification and read abundance to calculate the relative frequency of the organisms within a sample. While 16S is a reliable method for identifying the relative frequency of organisms it does not provide reliable functional information about the genes encoded by these organisms. As a consequence, whole-metagenome sequencing (WMS) data has been increasingly utilized with the goal of providing functional information about the organisms present. Most notably, WMS allows for an increased depth and specificity of sequenced species as well as insights into gene abundance and metabolic capacity. Yet, a limitation of WMS is the large number of sequence reads which must be mapped to databases, which requires significant expertise to balance classification accuracy with discarded reads. Our group previously compared data from

16s rRNA and Whole Genome Sequencing (WGS)⁴⁵. We found that diversity, evenness, and richness measures are comparable when using 16s rRNA and WGS, while relative abundances of individual genera differed between the two methods. Based on the comparisons, we found that 16s rRNA is more accurate in taxonomic classifications when looking at proportions and abundances.

-Edits to the manuscript can be found on page 12, lines 271-275

Edits read: “Our group previously compared data from 16s rRNA and Whole Genome Sequencing (WGS)⁴⁵. We found that diversity, evenness, and richness measures are comparable when using 16s rRNA and WGS, while relative abundances of individual genera differed between the two methods. Based on the comparisons, we found that 16s rRNA is more accurate in taxonomic classifications when looking at proportions and abundances.”

REVIEWER 1, COMMENT 4 (Results)

“This study also has a problem with respect to participant distributions. For example, supplemental Figure 1, you can find out such a small number of participants in histology, grade, and stage.”

Response:

-We recognize our studies' modest sample size remains a potential limitation. Yet despite the relatively small size of this prospective patient cohort, large statistically significant differences were still observed, alluding to the underlying complexity associated with the cervical microbiome in this patient population. We appreciate the reviewer's feedback and have included a comment in regards to this in the limitations paragraph of the text.

-Edits to the manuscript can be found on page 13, lines 277-281

Edits read: Consequently, the sample size limited our ability to weigh statistical power.

REVIEWER 1, COMMENT 5 (Results)

“In this study, many of the results showed trends of significance. The results should be explained very carefully.”

Response:

-We appreciate the reviewer’s feedback. We recognize that this is a limitation of the study and were careful to not over state our conclusions throughout the manuscript.

REVIEWER 1, COMMENT 1 (Discussion)

“Gut microbiome is associated with obesity based on previous literature. When explaining the obesity paradox, it brings out another question whether this gut microbiome difference was caused by obesity. Also, there is a conflict in relationships between BMI (continuous vs categorical variables) and survival outcomes, please explain this.”

Response:

-We appreciate the reviewer’s feedback. Our results suggest that overweight or obesity is a favorable prognostic factor independent of gut diversity. Unfortunately obesity was not the initial focus of this study. Our study design prevents us from understanding causal associations or mechanisms linking differences in the gut microbiota and obesity which is an area that deserves further study. However, we agree with the reviewer’s critique and determining the role obesity plays with respect to the gut microbiome in gynecologic malignancies would be of interest in future validation studies.

Differences in the relationship of survival outcomes and BMI when considered as a continuous variable vs. a categorical variable are mainly due to the way the hazard ratios are computed. In the case of a categorical variable, the hazard ratio is computed for each category with respect to a reference category. We had categorized BMI in order to visualize the relationship with OS and

RFS in a Kaplan Meier curve, however when considering a Cox proportional hazard model, it is always best to depict BMI as a continuous variable. Thus these differences are purely due to the nature of the variable.

REVIEWER 1, COMMENT 2 (Discussion)

“Results seem to miss the associations between the gut microbiome + immune cells and survival outcomes.”

Response:

-We appreciate the reviewer’s feedback. The association between gut diversity and local immune signatures suggest that patients with high gut diversity develop increased infiltration of activated CD4+ T-cell subsets. Higher alpha gut diversity at baseline correlated with an improved RFS and OS. Taken together, our results imply that the diversity of gut microbiota may contribute to increased infiltration of activated CD4+ T-cells which contribute to favorable responses to CRT treatment.

-Edits to the manuscript can be found on page 8, lines 179-181 and lines 182-184.

Edits read: “The association between gut diversity and local immune signatures suggest that patients with high gut diversity develop increased infiltration of activated CD4+ T-cell subsets and highlights helper CD4+ T cells as potential mediators of antitumor immunity upon CRT treatment. Taken together, our results imply that the diversity of gut microbiota and increased infiltration of activated CD4+ T-cells might be a shared benefit factor in those who respond well to CRT treatment.”

REVIEWER 2, MAJOR COMMENT 1

“The authors concluded that gut microbiome diversity is an independent predictor of survival in cervical cancer patients receiving chemoradiation, however this conclusion might have altered

depending on the predefined cutoff points of p-value for the survival analysis. The authors seem to defined significant p-value as <0.1. I understood the setting looser (than <0.05) p-value can be an option considering the relatively small number of patients in total, but this part needs to be rationalized to lead the firm conclusions as the above. At least, the authors should add careful explanations on the limitations of the study. Also, the author may be encouraged to soften the expression for the prognostic impact of microbiome in discussion section as well as the main title”

Response:

-We appreciate the reviewer’s feedback. For our general analysis, we had set our significance level to an alpha of 5%. However, in order to decide which variables should be included in the multivariate survival analysis, we used a p-value of 0.1 as a cut-off in the univariate analyses. We recognize that this might cause some confusion and have made necessary changes to make the distinction clearer.

-Edits to the manuscript can be found on page 17, lines 379-381.

Edits read: “A receiver operating characteristic (ROC) curve was calculated by plotting the values of sensitivity vs. 1-specificity as the value of the cut-off moves from 0 to 1. A table was constructed using the coordinates of the curve in order to identify the most appropriate cut-off for SDI in which the sensitivity and specificity are maximized. The cut-off for SDI was identified to be 2.69. Patient and tumor characteristics were analyzed by univariate and multivariate Cox regression models for Recurrence-free survival (RFS) and Overall survival (OS). Covariates with a p-value of less than or equal to 0.1 in the univariate model were chosen to be included in the multivariate analysis. Recurrence free survival (RFS) was defined as the time from the initiation of the therapy until the date of disease relapse or progression or death from any cause. Disease relapse or progression was defined as appearance or signs of the disease that was confirmed pathologically or

radiographically. Overall survival (OS) is defined as the time from the date of initiation of therapy until the date of death from any cause. Survival was estimated by the Kaplan–Meier method, and comparisons were made by log-rank tests. Characteristics included age, body mass index (BMI), race, FIGO stage, grade, histology, nodal status, smoking status, antibiotic use and max tumor size. For each outcome of interest, a multivariate Cox regression analysis was performed to adjust for the effects of prognostic factors identified on univariate analysis as influencing survival in cervical cancer. These analyses were conducted using covariates with $p \leq 0.1$ in a stepwise fashion.”

REVIEWER 2, MAJOR COMMENT 2

“The high percentage of antibiotics use could have impact on the results from this study. The authors should mention this in the discussion and provide the details (types etc.) of the antibiotics.”

Response:

-We recognize that this is a limitation of the study. Unfortunately antibiotic use was not the initial focus of this study and we have incomplete baseline data to expand our current analysis. All courses of antibiotics was given after baseline were obtained. However, we agree with the reviewer’s critique and have created and included a new supplemental table providing details of antibiotic use.

REVIEWER 2, MINOR COMMENT 1

“Page 6, line 116: There is a discordance between the cutoff of p-value of univariate analysis in result ($P \leq 0.2$) and Method (< 0.1). This would be critical and the authors should clarify which is the predefined p-value.”

Response:

-We agree with this clarification and have made necessary changes.

-Edits to the manuscript can be found on page 6, line 123.

Edits read: “In the univariate Cox proportional hazard regression model predicting RFS, 3 covariates showed $P < 0.1$.”

REVIEWER 2, MINOR COMMENT 2

“Page 6, line 132-134: The authors need to describe the methods of calculation by receiver operating characteristic curve. Also, the authors are encouraged to provide the ROC as a supplemental material.”

Response:

-We appreciate the reviewer’s feedback. We calculated a receiver operating characteristic (ROC) curve by plotting the values of sensitivity vs. 1-specificity as the value of the cut-off moves from 0 to 1. We then constructed a table using the coordinates of the curve in order to identify the most appropriate cut-off for SDI in which the sensitivity and specificity are maximized. The cut-off for SDI was identified to be 2.69.

-Edits to the manuscript can be found on page 16, lines 379-381.

Edits read: “A receiver operating characteristic (ROC) curve was calculated by plotting the values of sensitivity vs. 1-specificity as the value of the cut-off moves from 0 to 1. A table was constructed using the coordinates of the curve in order to identify the most appropriate cut-off for SDI in which the sensitivity and specificity are maximized. The cut-off for SDI was identified to be 2.69.”

REVIEWER 2, MINOR COMMENT 3

“Page 6, line 134-142: The authors should describe the survival differences referring to their statistical differences (just numerically or statistically) according to their predefined level, not just as “longer”.”

Response:

-We agree with this clarification and have made necessary changes. The manuscript now reads “The median RFS was 44 months for patients with low fecal alpha diversity at baseline, and not reached for patients with high fecal alpha diversity at baseline (P = 0.16). The median OS was 48 months for patients with low fecal alpha diversity at baseline, and again not reached for patients with high fecal alpha diversity at baseline (P = 0.094).”

-Edits to the manuscript can be found on page 7, line 142-145.

Edits read: “The median RFS was 44 months for patients with low fecal alpha diversity at baseline, and not reached for patients with high fecal alpha diversity at baseline (P = 0.16) (Fig. 1a). The median OS was 48 months for patients with low fecal alpha diversity at baseline, and again not reached for patients with high fecal alpha diversity at baseline (P = 0.094) (Fig. 1b).”

REVIEWER 2, MINOR COMMENT 4

“Page 7, line 155-157: The authors should describe the survival differences referring to their statistical differences (just numerically or statistically) according to their predefined level, not just as “prolonged”.

Response:

-We agree with this clarification and have made necessary changes. The manuscript now reads “In patients with a low relative abundance of *Veillonella* at baseline the median OS was 48 months, and in patients with a high relative abundance of *Veillonella* at baseline the median OS was still not reached (P = 0.054).”

-Edits to the manuscript can be found on page 8, line 165-167.

Edits reads: “In patients with a low relative abundance of *Veillonella* at baseline the median OS was 48 months, and in patients with a high relative abundance of *Veillonella* at baseline the median OS was still not reached (P = 0.054).”

REVIEWER 2, MINOR COMMENT 5

“Page 7, line 159-160: The following part should not be included in the result section: “Because the gut microbiota is thought to influence disease progression partially through modulating local immune response”. This information should be given in the intro section with appropriate references.”

Response:

-We appreciate the reviewer’s feedback. We agree with this clarification and have made necessary changes.

-Edits to the manuscript can be found on page 8, line 169-170

Edits read: “High SDI was positively correlated with tumor infiltration of CD4 T cells at week 3, and CD4ki67+ T-cells at week 5, (Table 3 and Fig. 3a-d).”

REVIEWER 2, MINOR COMMENT 6

“Page 7, line 160- page 8, line 163: The following part should not be included in the result section: “we analyzed the cervical tumors in our cohort of patients via flow cytometry on tumor brushings performed before week 1, week 3 and week 5 of radiation therapy. To identify features associated with high gut diversity, Spearman correlation analysis was conducted between immune signatures at each time point”. This information should be given in the method section.”

Response:

-We appreciate the reviewer’s feedback. We agree with this clarification and have made necessary changes.

-Edits to the manuscript can be found on page 16, line 357-359.

Edits read: “In order to explore the association between the gut microbiota and immune signatures we analyzed the cervical tumors in our cohort of patients via flow cytometry on tumor brushings performed before week 1, week 3 and week 5 of radiation therapy.”

REVIEWER 2, MINOR COMMENT 7

“Page 8, line 165-166: The following part should not be included in the result section: “These results suggest that patients with high gut diversity develop increased infiltration of activated CD4+ T-cell subsets”. This speculation should be provided in the discussion.”

Response:

-We appreciate the reviewer’s feedback. We agree with this clarification and have made necessary changes.

-Edits to the manuscript can be found on page 8, line 179-184

Edits read: “Lastly, the association between gut diversity and local immune signatures suggest that patients with high gut diversity develop increased infiltration of activated CD4+ T-cell subsets and highlights helper CD4+ T cells as potential mediators of antitumor immunity upon CRT treatment. Taken together, our results imply that the diversity of gut microbiota and increased infiltration of activated CD4+ T-cells might be a shared benefit factor in those who respond well to CRT treatment.”

REVIEWER 2, MINOR COMMENT 8

“Page 13, line13: Please provide the company and catalog number for the swab and lysis buffer the authors used in this study.”

Response:

-We appreciate the reviewer’s feedback. We agree with this clarification and have made necessary changes.

-Edits to the manuscript can be found on page 15, line 322-323.

Edits read: “(product # DSK-50 and XME-50, www.isoHelix.com, UKSamples)”

REVIEWER 2, MINOR COMMENT 9

“Figure 1: The authors are encouraged to add the number at risk below for each K-M curve. This would help the readers to easily grasp the sample size in each subgroup and more appropriate interpretation of the data. The cutoff of BMI should be stated in figure legends (or in figures).

“Diversity” should be replaced to “Baseline alpha-diversity”.”

Response:

-We appreciate the reviewer’s feedback. We agree with this clarification and have made necessary changes.

-Edits to the manuscript can be found on page 18, line 405, as well as in the figures. “Diversity” has been replaced with “Baseline alpha-diversity” throughout the manuscript.

Edits read: “BMI cut-offs were based on WHO standards”

REVIEWER 2, MINOR COMMENT 10

“Supplemental Fig 1: Diversity -> Alpha diversity”

Response:

-We appreciate the reviewer’s feedback. We agree with this clarification and have made necessary changes.

-“Diversity” has been replaced with “Alpha-diversity” throughout the manuscript.

REVIEWER 2, MINOR COMMENT 11

“Table1, 2: In the row of characteristics, please provide the number of patients categorized into each subgroup. Also, abbreviation should be provided (not only for CI and HR). For “significant P value”, the author should specify the cutoff value in the footnote of the table”

Response:

-We appreciate the reviewer's feedback. Patient numbers are provided in supplemental table 1. The footnotes' of Table 1 and 2 have been edited to include all abbreviations and the cutoff value has been specified for significant p-values.

REVIEWER 2, MINOR COMMENT 12

“Supplemental table 1: The percentages for the patients seem to be wrong in the subgroup of Race (Asian, Black). Please double check for the other subgroups.”

Response:

-We appreciate the reviewer's feedback. We agree with this clarification and have made necessary changes to Supplemental Table 1.

REVIEWER 3, COMMENT 1

“Thank you for the opportunity to review this manuscript. The authors have provided a clear and concise dataset further implicating the microbiota in treatment outcomes for cancer. They have specifically built upon existing research to demonstrate a specific microbial contribution to radiotherapy efficacy, an approach that to date has not been explored well. I believe these findings will be relevant to the broader research community, prompting new approaches aimed at modulating the microbiota to improve outcome.”

Response:

-We appreciate the reviewer's comments. We believe that the significance of our study lies in that modulation of gut microbiota before chemoradiation might provide an alternative way to enhance treatment efficacy and improve treatment outcomes in cervical cancer patients.

REVIEWER 3, COMMENT 2

“The authors have provided a convincing dataset despite the fairly low sample size, and their approach is complemented by immune analysis to provide mechanistic insight regarding how the microbiota confers its effect on treatment efficacy.”

Response:

-We appreciate the reviewer’s comments.

REVIEWER 3, COMMENT 3

“What is disappointing about this paper is the siloed approach to outcome, with survival discussed without any mention of toxicity. The microbiota is well documented to contribute to acute and chronic toxicities, particularly radiation-induced gastrointestinal mucositis/proctitis. The indirectly affect survival by influencing treatment compliance, dosing schedules, etc. Evidence in women with cervical cancer already shows that women who develop toxicity have vastly different microbial communities compared to those that do not develop toxicity. This has not been discussed in the paper at all and represents a key oversight in the approach used by authors, particularly given the overlap in toxicity and efficacy. I would recommend inclusion of toxicity data if it has been collected to determine if there is any influence of toxicity profiles on OS/PFS or if the same microbial findings can be applied to toxicity prediction. Alternatively, if this is not possible, the authors must address this in the discussion with clear reference to how their results can be applied to improve efficacy without inadvertently promoting toxicity.”

Response:

-We appreciate the reviewer’s comments. We agree with the reviewer in that studying toxicity alongside survival is of great importance. Our collaborative group has previously published on the association between gut microbiome changes and patient-reported GI toxicity through the course of radiation therapy in a cohort of women receiving radiation therapy for the definitive

management of cervical cancer. We found that over time increased radiation toxicity is associated with decreased gut microbiome diversity. Additionally baseline diversity was not predictive of end-of-treatment bowel toxicity, but gut composition may identify patients at risk for developing high toxicity. Our collaborative group does not have complete patient toxicity and outcome data available to present at this time for this cohort, but this data will be presented in a future separate report. We appreciate the reviewer's feedback and have included a comment in regards to this in the discussion section of the text.

-Edits to the manuscript can be found on page 11, lines 240-241.

Edits read: “Recently, our collaborative group investigated the association between patient reported bowel toxicity and the gut microbiota at baseline in thirty five patients receiving radiation therapy for the definitive management of cervical cancer³⁰. We found increased radiation toxicity to be associated with a decrease in gut microbiome diversity. Additionally, although baseline gut diversity was not predictive of end-of-treatment bowel toxicity, we identified compositional characteristics associated with members of the Clostridiales order in patients who experienced less toxicity. Finding ways to combat radiation associated toxicity is of great interest to overall patient well-being. For radiation GI toxicity that is not amendable to standard treatment, modulation of gut microbiota before chemoradiation might provide an alternative way to enhance not only treatment efficacy but improve treatment associated toxicity in cervical cancer patients.”

REVIEWER 3, COMMENT 4

“Furthermore, the authors do not discuss the specifics of their microbial findings. It seems counterintuitive that "pathogenic" strains like Shigella and Enterobacteriaceae would correlate with optimal survival - could the authors please expand on this finding and postulate why/how pathogenic bacteria may promote radioefficacy. Again, considering potential

immunosuppression in cancer patients, increasing the abundance of these microbes to boost efficacy may be undermined by heightened risk of infection.”

Response:

-We appreciate the reviewer’s comments. In our analysis long term survivor samples were significantly enriched in *Escherichia Shigella*, *Enterobacteriaceae*, and *Enterobacteriales*. Previous evidence suggests distinct gut organisms can enhance systemic and antitumor immune response. This is thought to be mediated by increased antigen presentation and improved effector T cell function in the tumor microenvironment which may modulate responses to treatment. We appreciate the reviewer’s feedback and have included a comment in regards to this in the discussion section of the text.

-Edits to the manuscript can be found on page 10, lines 225-230.

Edits read: “In our analysis long term survivor samples were significantly enriched in *Escherichia Shigella*, *Enterobacteriaceae*, and *Enterobacteriales*. Previous evidence suggests distinct gut organisms can enhance systemic and antitumor immune response. This is thought to be mediated by increased antigen presentation and improved effector T cell function in the tumor microenvironment which may modulate treatment outcomes¹¹.”

REVIEWER 3, COMMENT 6

“Overall, a solid study with some interesting findings likely to be relevant across many areas of cancer. I wish the authors all the best.”

Response:

-We appreciate the reviewer’s comments and thank them for their feedback.

Sincerely,

Ann H. Klopp, M.D., Ph.D.
Associate Professor
Dept. of Radiation Oncology
The University of Texas MD Anderson Cancer Center

REVIEWERS' COMMENTS:

Reviewer #1 (Remarks to the Author):

Thanks for the revision of this manuscript. I still have several comments:

1. Age was suggested as a significant factor in the univariate and multivariate analysis. However, discussion about it is missing completely from the whole pic. Also the age range of the participants is wide.
2. Several taxa were identified associated with the survival. More information about these taxa and how they potentially function to improve or exacerbate survivals should be discussed in kinda detail.
2. The results regarding associations between Gut Microbiota Profile and Immune Signatures in page 8 lines 168-170 seem not complete. A complete description of the results is needed.
3. Regarding the data collect, stool was collected by swabs. It is unclear whether it is rectal swabs or stool samples. Please clarify this in page 14 line 319.
4. There are some minor things to fix: a) page 6 line 120, Supplementary Figure 1a-e? b) page 14 line 319-321, stool specimen was collected at 5 timepoints. Please address that this study only used the baseline stool samples and make comments on Figure 1a as well.

Reviewer #2 (Remarks to the Author):

Thank you the point-by-point responses to my comments. Despite of several limitations of this research, it is interesnting and worth considering for publication in this journal.

Reviewer #3 (Remarks to the Author):

The authors have provided a comprehensive response to my queries/concerns raised in the initial review, whilst also addressing those of the other reviewers. I have no further comments.

Manuscript ID: COMMSBIO-20-1670-T

Title: Gut microbiome diversity is an independent predictor of survival in cervical cancer patients receiving chemoradiation

Dear Communications Biology Reviewers,

We are very excited to have been given a second opportunity to revise our manuscript entitled “Gut microbiome diversity is an independent predictor of survival in cervical cancer patients receiving chemoradiation” for Communications Biology. We want to extend our appreciation for taking the time and effort necessary to provide such insightful guidance. Herein, we explain how we revised the paper based on your comments and recommendations.

Note: Page and line numbers refer to the edited manuscript with revisions

REVIEWER 1, COMMENT 1

“Age was suggested as a significant factor in the univariate and multivariate analysis. However, discussion about it is missing completely from the whole pic. Also the age range of the participants is wide”

Response:

-We appreciate the reviewer’s feedback. In this study we found age to be an independent risk factor for RFS in cervical cancer patients undergoing chemoradiation. We previously characterized the fecal microbiome in women with cervical cancer and found the diversity of the fecal microbiome to differ between young and older women. Other have found advanced age to be associated with variations in the gut microbiota composition characterized by a loss of diversity of specific taxa.

Our finding of age as an independent risk factor for RFS in cervical cancer may signify a temporal association between an altered bacterial composition in the gut and response to therapy.

-Edits to the manuscript can be found on page 12, lines 272-278.

Edits read: “We found age to be an independent risk factor for RFS in cervical cancer patients undergoing chemoradiation. We previously characterized the fecal microbiome in women with cervical cancer and found the diversity of the fecal microbiome to differ between young and older women. Other have found advanced age to be associated with variations in the gut microbiota composition characterized by a loss of diversity of specific taxa. Our finding of age as an independent risk factor for RFS in cervical cancer may signify a temporal association between an altered bacterial composition in the gut and response to therapy”

REVIEWER 1, COMMENT 2

Several taxa were identified associated with the survival. More information about these taxa and how they potentially function to improve or exacerbate survivals should be discussed in kinda detail.”

Response:

- We appreciate the reviewer’s feedback. Specific taxa identified in our microbiome analysis is discussed in the discussion section on page 10, lines 230-235 and page 11, lines 248-250. We agree that additional mechanistic insights on the specific tax are needed, unfortunately this was not the initial focus of this study and we cannot expand our current analysis. However, we hope to they explore how specific taxa potentially function to improve or exacerbate survival in future validation studies.

REVIEWER 1, COMMENT 3

“The results regarding associations between Gut Microbiota Profile and Immune Signatures in page 8 lines 168-170 seem not complete. A complete description of the results is needed.”

Response:

- We appreciate the reviewer’s feedback. A more complete description of these results has been added.

- Edits to the manuscript can be found on page 8, lines 169-174

Edits read: “High SDI was positively correlated with tumor infiltration of CD4+CD69+ and CD4+PD1+T-cells at week 3, and CD4+ki67+ T-cells at week 5 ($P=0.004$, $Q=0.14$; $P=0.036$, $Q=0.21$; $P=0.004$, $Q=0.07$, respectively; Table 3). Testing for a correlation between Shannon diversity index and CD4+ showed no significant associations (Fig 3a). However, Spearman correlation showed a positive correlation between Shannon diversity index and CD4+ Ki67+, CD4+ CD69+, and CD4+ PD1+ with higher Shannon diversity correlating with higher T-cells (Fig. 3b-d).”

REVIEWER 1, COMMENT 4

“Regarding the data collect, stool was collected by swabs. It is unclear whether it is rectal swabs or stool samples. Please clarify this in page 14 line 319.

Response:

-We appreciate the reviewer’s feedback. Rectal swabs were collected from all patients by a clinician performing rectal exams at five time points (baseline; weeks 1, 3, and 5 of radiotherapy; and 3 months after CRT completion) using a matrix-designed quick-release Isohelix swab to characterize the diversity and composition of the microbiome over time (product # DSK-50 and

XME-50, www.isohelix.com, UKSamples). The swabs were stored in 20 µl of protease K and 400 µl of lysis buffer (Isohelix) and kept at -80°C within 1 h of sample collection

-Edits to the manuscript can be found on page 15, line 332

Edits read: “Rectal swabs were collected from all patients by a clinician performing rectal exams at five time points (baseline; weeks 1, 3, and 5 of radiotherapy; and 3 months after CRT completion) using a matrix-designed quick-release Isohelix swab to characterize the diversity and composition of the microbiome over time (product # DSK-50 and XME-50, www.isohelix.com, UKSamples).”

REVIEWER 1, COMMENT 5

“There are some minor things to fix: a) page 6 line 120, Supplementary Figure 1a-e? b) page 14 line 319-321, stool specimen was collected at 5 timepoints. Please address that this study only used the baseline stool samples and make comments on Figure 1a as well.”

Response:

- We agree with this clarification and have made necessary changes.

-Edit to the manuscript can be found on page 6, line 120 and page 25 line 595.

REVIEWER 2, COMMENT 1

“Thank you the point-by-point responses to my comments. Despite of several limitations of this research, it is interensting and worth considering for publication in this journal”

Response:

-We appreciate the reviewer’s comments and thank them for their feedback.

REVIEWER 3, COMMENT 1

“The authors have provided a comprehensive response to my queries/concerns raised in the initial review, whilst also addressing those of the other reviewers. I have no further comments.”

Response:

-We appreciate the reviewer's comments and thank them for their feedback.

Sincerely,

Ann H. Klopp, M.D., Ph.D.
Associate Professor
Dept. of Radiation Oncology
The University of Texas MD Anderson Cancer Center